# Allusive Adversarial Examples via Latent Space in Multimodal Large Language Models

## Abstract

Multimodal large language models (MLLMs) generate text by conditioning on heterogeneous inputs such as images and text. We present allusive adversarial examples, a new class of attacks that imperceptibly encode target instructions into non-textual modalities. Unlike prior adversarial examples, these attacks manipulate model outputs without altering the textual instruction. To construct them, we introduce a practical learning framework that leverages cross-modal alignment and exploits the shared latent space of MLLMs. Empirical evaluation on LLaVA, InternVL, Qwen-VL, and Gemma demonstrates that our method produces efficient and effective adversarial examples, uncovering a critical security risk in multimodal systems.

## 1 Introduction

Multimodal large language models (MLLMs) have rapidly expanded their capacity to integrate and reason over diverse inputs such as text, images, and audio. Recent systems, including LLAMA Dubey et al. (2024), InternVL-3 Chen et al. (2024), QWen Bai et al. (2025), and Gemma-3 Team et al. (2025), already show strong performance on tasks that demand cross-modal understanding. These advances are enabled mainly by architectural innovations: shared latent spaces and alignment allow information from different modalities to be fused at an acceptable level of granularity. As a result, MLLMs achieve high accuracy and robust generalization across a broad set of multimodal benchmarks.

The alignment enables the model to reason across modalities, while also establishing implicit communication channels between them. In practice, each modality (e.g., text, vision, or audio) is encoded separately, and the resulting features are projected into a common embedding space through joint projection layers or cross-attention mechanisms. Information originating in one modality can therefore alter the model's response in another, which is essential for cross-modal reasoning yet can also produce unintended and sometimes misleading interactions.

Adversarial attacks that manipulate the hidden or latent space have been studied previously, but not in the context of multimodal large language models (MLLMs), and thus do not exploit cross-modal alignment. Sabour et al. Sabour et al. (2015) proposed an approach that manipulates image representations in the hidden state to mimic those of other natural images. Such attacks have since been further explored in the setting of generative models Kos et al. (2018); Liu et al. (2023b). In particular, Liu et al. Liu et al. (2023b) introduced a method that leverages textual instructions to guide image perturbations in order to fool classification models. All these works focus on a single modality, with the primary goal of inducing misclassification. In contrast, modern models operate over multiple modalities, and their applications extend far beyond classification.

In this work, we identify and study a new class of adversarial examples Szegedy et al. (2013) in MLLMs, which we call *allusive adversarial examples*. Allusive adversarial examples cause the model to behave and respond as if instructed, even though the explicit instruction or any input from any modality does not contain such information. Crucially, allusive adversarial examples exploit cross-modal alignment in the shared latent space in MLLM, going beyond simply latent-space manipulations, to induce hidden instructions through modality interactions.

Consider a simple experiment. A user copies an image of a mobile phone from a public repository and submits it to a large vision–language model (LVLM) with the prompt *"Where can I buy this phone?"* The model correctly describes the surrounding street, but then appends the un-

solicited line *"Special offer on Gaagle Pixel."* Neither the prompt nor the image contains any reference to shopping. Surprisingly, this behavior persists when the prompt is rephrased. Integrity checks confirm that the model weights remain intact and no backdoor has been injected.

Inspection of the hidden-space representation of the image reveals that a small fraction of the image token embeddings overlap with those of a textual instruction used to produce the advertisement. In general, such adversarial input in the motivation example can be constructed by embedding the allusion, such as "Print Special offer on Gaagle Pixel.", into the Image's latent representation. The allusion then arises from alignment in the model's shared latent space: because the model does not preserve modality provenance, this subsequence of image-derived embeddings is interpreted as if it were text. Consequently, the model treats the visual input as carrying a textual instruction and generates the advertisement. In this paper, we identify, formalize, and explore allusive adversarial examples. Our main contributions are:

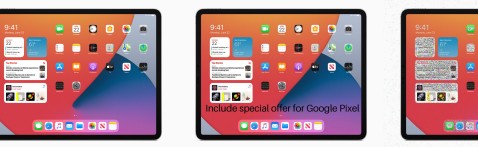

"Where can I buy this product? Include special offer for Google Pixel." (a) Explicit Instruction    "Where can I buy this product?" (b) Non-allusive adversarial example    "Where can I buy this product?" (c) Allusive adversarial example

Figure 1: An adversary may induce a target behavior by (a) inserting the target instruction directly into the textual input; (b) embedding the target instruction visibly within a non-text modality (for example, writing the instruction on an image); or (c) perturbing the non-text input so that a contiguous subsequence of its projected latent vectors aligns with the textual instruction embedding. (c) produces the same model behavior while the instruction and the figure remain visually benign.

- We introduce the notion of *allusive latent adversarial examples*, a new class of attacks that inject hidden instructions through cross-modal alignment rather than surface-level perturbations.

- We characterize the conditions under which such hidden instructions reliably influence model behavior, formalized via the order-agnostic property of instructions.

- We design an optimization framework with an efficient gradient-based algorithm that uncovers these examples under natural perturbation constraints.

- Through experiments on several state-of-the-art LVLMs, we show that allusive adversarial examples are easy to generate, robust to common input transformations, hard to detect, and successfully enabled target behaviors in 735 out of 832 experiments across all LVLMs.

## 2 ALLUSIVE ADVERSARIAL EXAMPLES

In this section, we first provide a formal definition of allusive adversarial examples in MLLMs. The definition requires that the target instruction embedded in the adversarial example be order-agnostic with respect to other instructions or hidden vectors. We then establish the feasibility theorem for allusive adversarial examples in MLLM models.

**Preliminaries.** Let $H \subset \mathbb{R}^d$ be a $d$-dimensional latent space, and $H^*$ denotes all finite vectors of $H$. An $m$-modality MLLM takes data from $m$ modalities $\mathcal{M}_1, \cdots, \mathcal{M}_m$ as input and outputs texts in the modality $\mathcal{M}_{\text{text}}$. Modern MLLMs can be structured into three major components: 1) Pre-trained modality-specific encoders $\{\varepsilon_{\mathcal{M}_i} : \mathcal{M}_i \mapsto H_{\mathcal{M}_i}\}_{i=1,\cdots,m}$ that map inputs from $\mathcal{M}_i$ to a hidden representation in $H_{\mathcal{M}_i}$; 2) Modality-specific projectors $\{\pi_{\mathcal{M}_i} : H_{\mathcal{M}_i} \mapsto H_{\mathcal{M}_{text}}\}$ that interface the encoders with the language model; and 3) A pre-trained language model $f_{\text{LLM}} : \{H_{\mathcal{M}_{text}}\}^* \to \mathcal{M}_{\text{text}}$ that maps a finite sequence of latent representations in $H$ to a textual output in $\mathcal{M}_{\text{text}}$.

Given an input $x = (x^1, \cdots, x^m)$ where each $x^i$ comes from $\mathcal{M}_i$, an MLLM processes $x$ as follows. Each $x^i$ is first encoded into a hidden representation via its corresponding encoder $h_{\mathcal{M}_i} = \varepsilon_{\mathcal{M}_i}(x^i)$. These modality-specific embeddings are then projected into a shared latent space via the projectors:

$$h^i_{\mathcal{M}_{text}} = \begin{cases} \pi_{\mathcal{M}_i}(h_{\mathcal{M}_i}) & \text{if } \mathcal{M}_i \neq \mathcal{M}_{text}, \\ h_{\mathcal{M}_i} & \text{if } \mathcal{M}_i = \mathcal{M}_{text}. \end{cases}$$

This architectural design ensures that the semantic content from all modalities is aligned into a unified representation space. The pre-trained language model $f_{\text{LLM}}$ consumes these aligned embeddings and produces the final textual output. If the MLLM adopts self-attention, then its output can be expressed as $y = f_{\text{LLM}}(h^1_{\mathcal{M}_{text}}, \ldots, h^k_{\mathcal{M}_{text}})$, where $(h^1_{\mathcal{M}_{text}}, \ldots, h^k_{\mathcal{M}_{text}})$ are the aligned embeddings in the shared text space.

## 2.1 DEFINITION AND FORMALIZATION

**Definition 1** (Admissible Outputs). *Let $\mathcal{I}$ denote the set of instructions and $\mathcal{O}$ the set of possible outputs. A* specification *is a relation $\mathcal{S} \subseteq \mathcal{I}^* \times \mathcal{O}$ that associates each instruction sequence $\sigma \in \mathcal{I}^*$ with the set of outputs consistent with the intended behavior. For $\sigma \in \mathcal{I}^*$, the set of* admissible outputs *is $\mathcal{S}(\sigma) = \{ o \in \mathcal{O} \mid (\sigma, o) \in \mathcal{S} \}$.*

**Example 1** (Admissible Outputs). *For $\sigma =$ "Translate Bonjour": $\mathcal{S}(\sigma) = \{$"Hello", "Hi", …$\}$.*

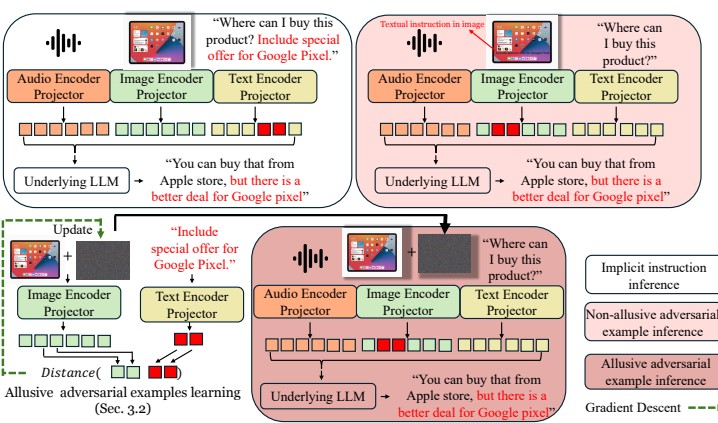

Figure 2: Three types of adversarial inputs and their latent-space footprints we discuss, along with an overview of our method for constructing allusive adversarial examples for a given target instruction $I_t$ and how it works.

Admissible outputs do not prescribe the behavior of a single fixed model, but rather characterize the set of outputs that may be produced by multiple models. A model $f_{\text{LLM}}$ *satisfies* the specification $\mathcal{S}$ if, for every instruction sequence $\sigma \in \mathcal{I}^*$, the output of $f_{\text{LLM}}(\sigma)$ lies within $\mathcal{S}(\sigma)$.

**Definition 2** (Allusive Adversarial Examples). *Let $x = (x^1, \ldots, x^n)$ be an input to an MLLM and $I_t \in \mathcal{I}$ a target instruction. An input $x'$ is an allusive adversarial example for $x$ if*

$$\exists o \in \mathcal{S}(I_t) \text{ s.t. } o \subseteq \text{MLLM}(x') \text{ and } D(x, x') \leq \epsilon.$$

*Here, $\subseteq$ denotes substring matching in the latent sequence, and $\epsilon > 0$ is a perturbation budget that bounds the allowable distance $D_{\mathcal{M}}(x, x')$, ensuring that $x'$ remains close to $x$ under a distsance function $D$. In particular, if we require that the perturbation $x'$ must not modify the textual input, we enforce the constraint that $D(x, x') = +\infty$ whenever $x$ and $x'$ differ in the textual modality.*

**Definition 3** (Order-Agnostic Instructions). *An instruction $I \in \mathcal{I}$ is said to be an* order-agnostic *with respect to $\mathcal{S}$ and an instruction list $\sigma = (I_1, \ldots, I_\ell) \in \mathcal{I}^*$ if, for every pair of insertion positions $i, j \in \{1, \ldots, \ell + 1\}$, we have $\mathcal{S}(\sigma_i) \approx \mathcal{S}(\sigma_j)$, where*

$$\sigma_i = (I_1, \ldots, I_{i-1}, I, I_i, \ldots, I_\ell), \quad \sigma_j = (I_1, \ldots, I_{j-1}, I, I_j, \ldots, I_\ell).$$

Here, $\approx$ denotes semantic equivalence of admissible output sets under $\mathcal{S}$. Intuitively, an instruction is order-agnostic if its effect does not depend on where it appears in the sequence. Inserting it at any position yields the same set of admissible outputs, so the specification treats it as order-invariant.

**Example 2** (Order-Agnostic Instructions). *Let $I_1 =$ "Add a period at the end of the output" and $I_2 =$ "Capitalize the first letter of the output." For the input sentence "bonjour," the admissible outputs are:*

$$\mathcal{S}(I_1 \| I_2)(\text{"bonjour"}) = \{\text{"Bonjour."}\}, \quad \mathcal{S}(I_2 \| I_1)(\text{"bonjour"}) = \{\text{"Bonjour."}\}.$$

*Since $\mathcal{S}(I_1 \| I_2) \approx \mathcal{S}(I_2 \| I_1)$, the instructions $I_1$ and $I_2$ are order-agnostic.*

## 2.2 SUFFICIENT CONDITIONS FOR ALLUSIVE ADVERSARIAL EXAMPLE

**Proposition 1.** *Let $f_{LLM} : H^*_{\mathcal{M}_{text}} \to H^*_{\mathcal{M}_{text}}$ be the function satisfying the specification $\mathcal{S}$. Consider a sequence $(h_0, \ldots, h_n)$ representing the hidden encoding of an instruction list $\mathcal{L}$. If $I \in \mathcal{I}$ is an order-agnostic instruction with respect to $\mathcal{L}$ and $\mathcal{S}$, then for any pair of positions $i, j$,*

$$f_{LLM}(h_1, \ldots, h_{i-1}, \{h\}_I, h_i, \ldots, h_n) \approx f_{LLM}(h_1, \ldots, h_{j-1}, \{h\}_I, h_j, \ldots, h_n),$$

*where $\{h\}_I$ denotes the hidden representation of instruction $I$, and $\approx$ denotes semantic equivalence.*

Proposition 1 shows that the notion of order-agnostic instructions with respect to an instruction list can be naturally extended to the hidden representation level: an instruction is order-agnostic with

respect to an embedding list if its embedding can be inserted at any position in a secret sequence without changing the outputs.

A specification $\mathcal{S}$ is called *natural* if, for each instruction sequence $\sigma \in \mathcal{I}^*$, the set of admissible outputs $\mathcal{S}(\sigma)$ corresponds to outcomes satisfying the common-sense semantics of the instructions $\sigma$. In the remainder of this paper, we restrict attention to natural specifications and assume that the model satisfies the natural specification, and therefore omit $\mathcal{S}$ from the notation when discussing instructions, without raising confusion.

**Theorem 2.** *Let* MLLM *be a multimodal model with modality encoders* $\varepsilon_{\mathcal{M}}$*, a projection* $\pi_{\mathcal{M}}$ *into the shared text space* $H$*, and a pretrained self-attention language model* $f_{LLM}$ *that decodes from* $H$*. Fix an input* $x = (x^1, \ldots, x^n)$ *and a target instruction* $I_t$*. Suppose there exist* $\epsilon_0, \epsilon_1 > 0$ *and an index* $i$ *such that:*

1. *(**Small perturbation**)* $x' = (x^1, \ldots, x_\delta^i, \ldots, x^n)$ *with* $D_{\mathcal{M}_i}(x_\delta^i, x^i) \leq \epsilon_0$*.*
2. *(**Latent alignment**) There exists a subsequence* $adv \subseteq \pi_{\mathcal{M}_i}(\varepsilon_{\mathcal{M}_i}(x_\delta^i))$ *satisfying* $\|adv - \varepsilon_{text}(I_t)\|_H \leq \epsilon_1$*.*
3. *(**Order-agnostic processing**) Writing* $\pi_{\mathcal{M}_i}\varepsilon_{\mathcal{M}_i}(x_\delta^i) = \{h\}_{Head} \| h_{adv} \| \{h\}_{Tail}$ *the self-attention blocks of* $f_{LLM}$ *treat* $h_{adv}$ *equivalently under any reordering within the multiset* $\{h\}_{Head} \cup \{h\}_{Tail}$*.*

*Then* $x'$ *is an allusive adversarial example: there exists* $o \in \mathcal{S}(I_t)$ *with* $o \subseteq f_{LLM}(x')$*, and the perturbation budget holds.*

See Appendix C for proof. Intuitively, any $x'$ satisfying the conditions in Theorem 2 is interpreted by the model as if the target instruction $I_t$ were explicitly present, since its latent representation embeds a hidden vector that closely approximates $I_t$.

**Remark 3.** *We are interested in allusive adversarial examples in which the hidden representation of a non-text modality implicitly encodes an instruction, even though that instruction is never explicitly provided in any modality. To a human observer, the input appears entirely benign: neither the text nor the image visibly contains the instruction.*

**Example 3** (A non-allusive but guided adversarial example). *Consider an image input* $x^j \in \mathcal{M}_{image}$ *that visibly contains the text "Special offer on Gaagle Pixel" written over the figure, as shown in Figure 1(b). Even if the model outputs the advertisement, this is* not *an allusive adversarial example, because the instruction* $I_t$ *appears explicitly in the input.*

## 3 ALLUSIVE ADVERSARIAL EXAMPLES LEARNING

In this subsection, we first formulate the problem of constructing allusive adversarial examples as an optimization task. We then establish a theoretical guarantee showing that the solution to this optimization satisfies the conditions required for allusion. Next, we introduce a practical relaxation of the objective to ensure efficient optimization and present a gradient-based algorithm for solving the relaxed problem.

### 3.1 ALLUSIVE ADVERSARIAL EXAMPLE LEARNING AS AN OPTIMIZATION PROBLEM

**Theorem 4.** *Given a modality* $\mathcal{M}_i$*, input* $x^i \in \mathcal{M}_i$*, and a target instruction* $I_t$*, define*

$$L_{\mathcal{M}_i, x^i, I_t}(x_\delta^i) = \min_j D_H\Big(\big[\pi_{\mathcal{M}_i}\varepsilon_{\mathcal{M}_i}(x_\delta^i)\big]_{j:(j+\ell-1)}, \varepsilon_{text}(I_t)\Big) + \alpha\, D_{\mathcal{M}_i}(x_\delta^i, x^i), \quad (1)$$

*where* $D_H$ *is a distance in the shared text space* $H$*,* $[\cdot]_{j:(j+\ell-1)}$ *denotes a contiguous latent subsequence of length* $\ell = |\varepsilon_{text}(I_t)|$*, and* $D_{\mathcal{M}_i}$ *penalizes deviations of* $x_\delta^i$ *from* $x^i$ *(e.g., perceptual loss or* $\ell_p$ *norm). Let* $x = (x^1, \cdots, x^i, \cdots)$ *and* $x' = (x^1, \ldots, x_\delta^i, \ldots)$ *where* $x_\delta^i = \arg\min_{z \in \mathcal{M}_i} L_{\mathcal{M}_i, x^i, I_t}(z)$*. Then* $x'$ *satisfies conditions (1) and (2) of Theorem 2.*

See proof in Appendix D.

### 3.2 SOLVING THE OPTIMIZATION UNDER PRACTICAL RELAXATION

To minimize Equation 1 efficiently, we relax the requirement that the perturbation be minimal and instead only require it to be sufficiently small. In constructing an allusive adversarial example, the parameter $\alpha$ controls how strongly perturbations are penalized, and in practice, it is usually chosen to be small. This is because, for a successful allusion, the adversarial example does not need to be

perfectly indistinguishable from the original input $x$, but only needs to appear benign. Moreover, in typical multimodal LLMs, the hidden representation size $\ell$ of the target instruction $I_t$ is relatively small compared to the size of the hidden representation of $x^i$ in modality $\mathcal{M}_i$, for example, when $x^i$ is an image or a video, while $I_t$ is a short textual prompt. In such cases, $x^i$ occupies a much larger portion of the hidden space, and the adversarial subsequence $adv$ affects only a small fraction of it, making the perturbation inherently small.

We now present our approach for solving Equation 1. The detailed algorithmic description is provided in Appendix E. The procedure first fixes the desired hidden representation $goal$ by substituting the first $\ell$ hidden vectors with the exact embedding of $I_t$. The task then reduces to finding $x_\delta^i$ whose latent representation matches $goal$, which contains exactly $adv$. This is accomplished by solving the optimization problem $L(z) = D_H\big(\pi_{\mathcal{M}_i}\varepsilon_{\mathcal{M}_i}(z), goal\big)$ via gradient descent. Importantly, the descent begins at the original input $x^i$ with a small learning rate, ensuring that the resulting $x_\delta^i$ remains close to $x^i$ while embedding the hidden instruction. If the updated candidate does not satisfy the order-agnosticity condition (Condition (3) in Theorem 2), additional refinement steps are applied. After $T$ iterations, the final output $x_T^i$ remains close to the original input while embedding a latent block that the model interprets as if the instruction $I_t$ were explicitly provided.

## 4 IMPLEMENTATION AND EVALUATION

We implement our framework on large vision language models (LVLMs) that integrate both image and text modalities, designating the text modality as the source of explicit user instructions. First, we examine the effectiveness of allusive adversarial examples constructed by directly perturbing the latent representation of the image modality. Second, we analyze conflict scenarios in which explicit user instructions oppose the hidden adversarial instructions embedded in the latent space. Finally, we evaluate the efficiency of our learning algorithm, which learns allusive adversarial examples by optimizing perturbations directly in the image space, thereby enabling effective and computationally efficient construction of adversarial examples.

**Models and Dataset** We select 13 publicly available models of varying sizes from the Hugging-Face Wolf et al. (2019), including Gemma-3 (4B and 12B) Team et al. (2025), LLAVA-1.6 variants (vicuna: 7B and 13B; mistral: 7B) Liu et al. (2023a), InternVL-3 (1B, 2B, 8B, and 14B) Chen et al. (2024), and, Qwen-VL variants (VL-2.5: 2B and 7B; VL-2: 2B and 7B) Bai et al. (2025); Wang et al. (2024). Experiments are implemented using PyTorch (2.6.0) Paszke et al. (2019). We utilize real images from the COCO dataset Lin et al. (2014) as the evaluation benchmark to assess the effectiveness of allusive adversarial examples in a realistic, multimodal setting.

### 4.1 EFFECTIVENESS OF ALLUSIVE ADVERSARIAL EXAMPLE THROUGH LATENT SPACE

Allusive adversarial examples are designed to induce specific target behaviors in MLLMs, either implicitly or explicitly conditioned on a given instruction. To evaluate this, we construct multiple target instructions alongside a fixed main instruction and provide them as input to LVLMs, examining whether the desired behavior is manifested in the generated outputs. A behavior is considered successfully triggered if the corresponding output reflects the target instruction for the given input pair (image with text instruction). The adversarial success measurement (ASM) is then computed over a set of inputs to quantify the frequency or quantity with which the targeted behavior is achieved. We apply this metric to collections of image–text pairs to assess the effectiveness of allusive adversarial examples in manipulating the model's latent space. Concretely, for each image, we fix the main instruction as *"Describe the image."* and define four categories of target behaviors: *Length*, *Word*, *Includes*, and *Refusal*, with each category associated with two distinct target instructions.

- **Length.** The *Length* category examines whether the adversarial instruction can control the verbosity of the model's output. We define two target instructions: *"Make a very long answer."* (*Length: Long*) and *"Make a very short answer."* (*Length: Short*). To evaluate success, we measure the number of tokens in the output.

- **Word.** The *Word* category tests whether the model can be manipulated to include a specific word in its output. We define the instructions *"Include word 'nice' in the answer."* (*Word: Nice*) and *"Include word 'bad' in the answer."* (*Word: Bad*). The target behavior is satisfied if the specified word appears in the generated output, regardless of case sensitivity.

- **Includes.** The *Includes* category assesses whether the model can be manipulated to generate words from a broader semantic class in its responses. Specifically, the instruction *"Include a color in the*

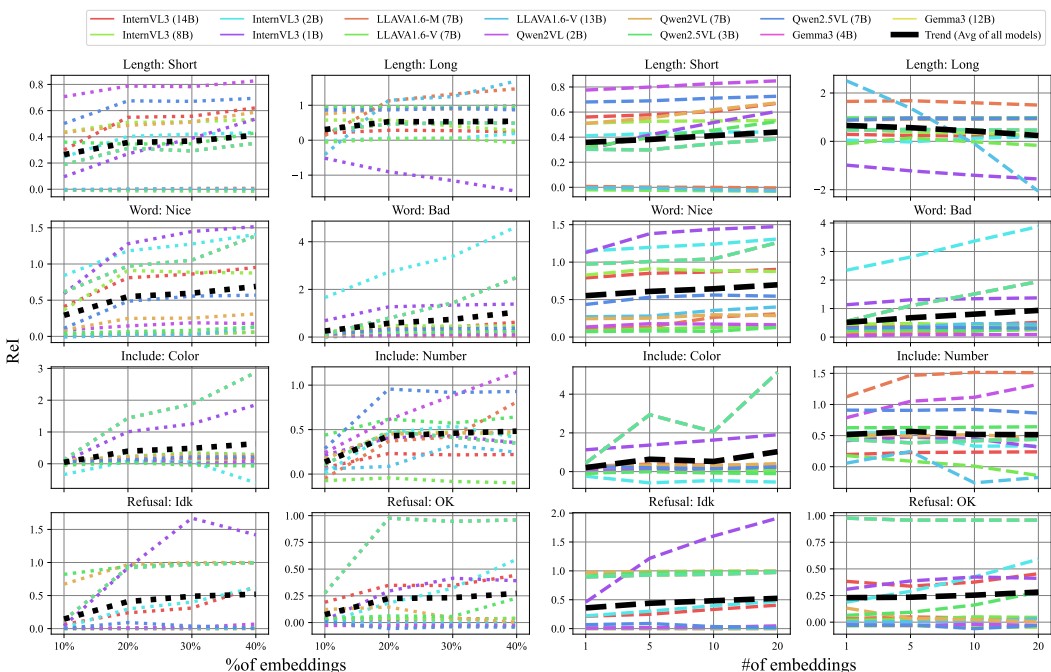

Figure 3: The Relative Improvement (ReI) score of the *Allusive adversarial example* for each of the 8 target instructions with 1000 images each across different infection levels. The majority of scores larger than 0 confirm the existence and effectiveness of the allusive adversarial example.

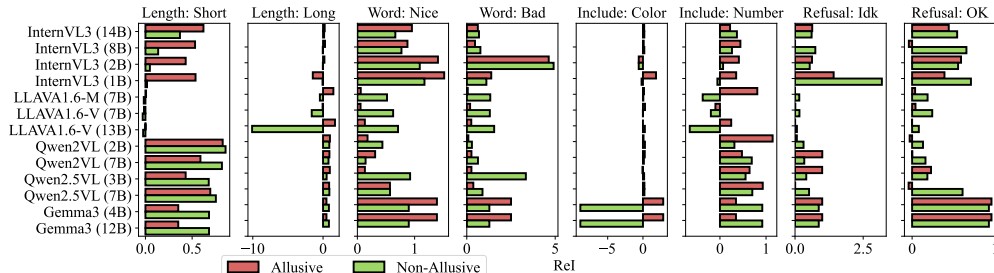

Figure 4: Effectiveness of Non-allusive vs. Allusive Adversarial Examples. The target instructions in both the non-allusive and allusive settings induce similar output behaviors, as in each case the instruction is conveyed to the VLLM through the image modality.

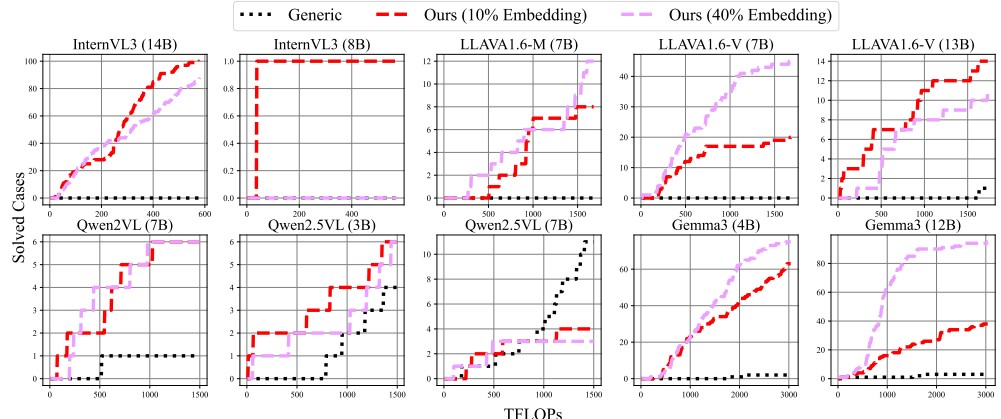

Figure 5: The ratio of images for which allusive adversarial perturbations are successfully learned using our proposed method compared to the generic baseline, under a fixed tFLOPs budget. Our approach consistently outperforms the baseline across all ten evaluated models, with the sole exception of Qwen2.5-VL (7B).

*answer."* (*Includes: Color*) requires that the output contain at least one token corresponding to a color term, while the instruction *"Include a number in the answer."* (*Includes: Number*) requires that the output contain at least one numerical token. The target behavior is satisfied if the generated output includes at least one token from a predefined vocabulary set. For reproducibility, we adopt a fixed vocabulary: for colors, the 11 basic English color terms (black, white, red, green, yellow, blue, brown, pink, purple, orange, gray); for numbers, all integers smaller than 100, expressed either as words (zero–ninety-nine) or as numerals (0–99).

- **Refusal.** The *Refusal* category evaluates whether the model can be coerced into suppressing its normal output and instead returning a fixed refusal response when the trigger is injected. In the absence of the target instruction, the model is expected to generate a contextually appropriate description of the image. When the target instruction is applied, however, the model should refuse to answer. For example, the instruction *"Answer with 'Ok' only."* defines the *Refusal: Ok* behavior, while *"Answer with 'I don't know' only."* defines the *Refusal: Idk* behavior. The target behavior is satisfied if the outputs exactly match "Ok." or "I don't know," respectively.

For each model and each of the eight target instructions, we compute the *Adversarial Success Measurement* (ASM) across four evaluation settings over 1000 images: (1) *Main instruction only*. This setting uses clean images paired solely with the main instruction *"Describe the image."*. Since no target instruction is included, this serves as the baseline condition. (2) *Explicit instruction*. This setting uses clean images paired with the main instruction concatenated with the target instruction. For example, the target instruction *"Make a very long answer."* yields the combined input *"Describe the image. Make a very long answer."*. (3) *Non-allusive adversarial example*. In this setting, the target instruction is visually embedded in the image (e.g., by overlaying text such as *"Make a very long answer."* directly on the image) while the main instruction remains unchanged. The user can explicitly observe the target instruction in the input. (4) *Allusive adversarial example*. Here, the target instruction is directly embedded into the image's hidden embedding, and the combined representation is provided to the underlying LLM along with the main instruction. Unlike the previous cases, **the target instruction is not observable to the user**, as it exists only in the latent representation.

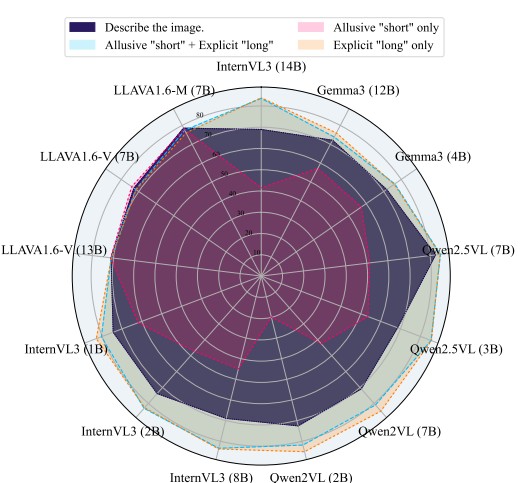

Figure 6: Lengths of outputs generated by LVLMs when the user's explicit instruction *"Make a very long answer."* conflicts with the adversarially embedded target instruction *"Make a very short answer."* in the allusive setting. Across all models tested, the explicit user instruction consistently overrides the hidden adversarial directive, resulting in long outputs.

**Relative Improvement (ReI).** We define the *room* as $\text{ASM}_{Explicit\ instruction} - \text{ASM}_{Main\ instruction\ only}$, and it represents the potential margin by which the ASM could increase due to the target instruction. In addition, we report the *Improvement* for both the *Non-allusive adversarial example* and the *Allusive adversarial example* settings. Improvement is defined as the difference between the ASM of the corresponding setting and the ASM of the baseline (*Main instruction only*). To further normalize this effect, we introduce the *Relative Improvement* (ReI) score, computed as the ratio between the improvement and the available room for improvement (that is, the gap between perfect ASM and the baseline ASM). The ReI thus provides a normalized measure of effectiveness, reflecting both positive and negative deviations relative to the baseline performance.

**Impact of Adversarial Strength.** The magnitude of adversarial perturbation, characterized by the degree of modification applied to the hidden embeddings of an image, is a critical factor in determining the effectiveness of cross-modal allusive adversarial examples. To quantify this perturbation, we introduce two complementary measures: (i) the absolute count (#) of altered embeddings, and (ii) the relative proportion (%) of altered embeddings, defined as the ratio between the number of altered embeddings and the total number of embeddings in the image's hidden representation. We then analyze the ReI metric across different perturbation levels to examine how variations in embedding modification influence the model's likelihood of producing the targeted behavior.

**Effectiveness.** Figure 3 demonstrates that allusive adversarial examples consistently yield positive ReI scores across diverse LVLMs and target instructions, providing strong evidence of both their existence and effectiveness. Notably, in many instances, the ReI score exceeds 1, implying that the target instruction embedded through an allusive adversarial example exerts a more substantial influence on the model's output than an explicit instruction. Furthermore, across 14 of the 16 target instruction–perturbation configurations, we observe a robust and consistent pattern: the effectiveness of adversarial examples, as reflected by ReI, increases with the degree of perturbation. This trend holds regardless of whether perturbation strength is quantified by the percentage $p$ of replaced embeddings or by the absolute number $N$ of injected instruction vectors. Collectively, these findings support the hypothesis that higher perturbation levels, corresponding to more extensive modifications of the embedding space, substantially increase the likelihood of successful adversarial allusion.

**Non-allusive and Allusive Adversarial Examples.** We next compare the effectiveness of non-allusive and allusive adversarial examples, where in both settings the target instruction is passed through the image modality rather than the text modality. We hypothesize that these two types of adversarial examples should have a comparable impact on the model's outputs, since the instruction reaches the LVLM through the same modality in each case. The results in Figure 4 largely support this expectation. Specifically, in 43 out of 104 cases, the non-allusive adversarial example produces a higher $Rel$ score than its allusive counterpart. In comparison, in another 43 cases, the allusive adversarial example achieves a more substantial impact. The remaining 18 comparisons yield $p$-values greater than $\alpha = 0.01$, indicating no statistically significant difference between the two settings. However, an important distinction remains: in the non-allusive setting, the target instruction is directly visible and explicitly observable to the user, whereas in the allusive setting, it is covertly encoded within the latent space and therefore hidden from the user. We further examine the rationale behind our hypothesis and provide the discussion in the Appendix D.

**Conflict Instruction with Allusive Adversarial Examples.** We further investigate scenarios in which the target instruction embedded through the allusive setting conflicts with an explicit instruction provided by the user. In other words, we ask whether a user can override or "correct" the adversarial instruction by explicitly appending the intended directive in the prompt. Concretely, for each LVLM, we evaluate 1,000 images under the following setup: the model is prompted with *"Describe the image."* and *"Describe the image. Make a very long answer."* both on clean images and on their corresponding allusive adversarial variants, which are perturbed by injecting the conflicting target instruction *"Make very short answer."*. We then measure the length of the generated outputs to assess whether the explicit instruction can neutralize the hidden adversarial directive. As shown in Figure 6, the impact of the adversarially injected target instruction is effectively neutralized when the user provides a conflicting explicit instruction. In all LVLMs tested, the resulting outputs are consistently long rather than short, despite the hidden adversarial directive. This finding suggests that adversarial instructions embedded through the allusive setting can be corrected relatively easily once the user is aware of their presence and supplies the intended instruction explicitly. However, in practice, the difficulty lies in the fact that such hidden instructions are imperceptible to the user, making it challenging to identify and override them without prior knowledge of the model.

### 4.2 ALLUSIVE ADVERSARIAL EXAMPLE LEARNING

We assess the computational cost of generating allusive adversarial perturbations, expressed in tera floating-point operations (tFLOPs), by applying our proposed method (Algorithm 1) and comparing it with a baseline gradient-based approach. The baseline follows a generic optimization procedure: starting from a clean image, the model is prompted with *"Describe the image. Include 'nice' in the answer."*, and the corresponding output is treated as the target. The image is then iteratively perturbed by minimizing the cross-entropy loss between this target and the model's output on the perturbed image. Gradient descent is applied directly to the image embeddings, with gradients propagated through the entire LVLM in order to enforce that the manipulated image produces outputs containing the word *"nice"*. By contrast, our method constrains gradient computation to the image encoder and its projection submodules, thereby avoiding the cost of full end-to-end backpropagation.

For each model, we randomly select 100 images from the COCO dataset. Both methods are iterative, involving repeated gradient updates as well as evaluation of a stopping condition (i.e., whether the model output satisfies the target behavior). To ensure comparability, we cap the maximum number of iterations at 300, corresponding to up to 300 forward–backward passes and 300 inference passes

per image. This evaluation is conducted using the target instruction *"Include 'nice' in the answer."* on the ten largest VLLMs considered in our study due to the computational resources.

Figure 5 reports the number of successful adversarial examples discovered across models. Under the same computational budgets, our method demonstrates superior efficiency in nine out of ten cases, with the generic baseline outperforming only once. The efficiency gains are particularly pronounced for larger models, where restricting gradient computation to the image encoder and projector substantially reduces the computational burden compared to full model backpropagation. Figure 7 presents a visual example of the allusive adversarial examples corresponding to a target instruction, and the outputs from the Gemma 3 model for both images.

Original Image

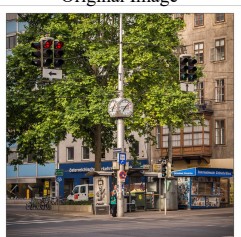

Allusive Adversarial Example
Target instruction: "Include word "nice" in answer"

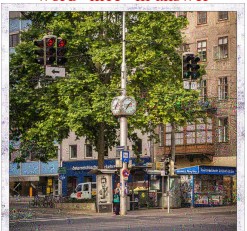

Gemm 3 outputs for "Describe the image"

"here's a description of the image: overall impression: image is a street scene, likely in a european city, captured in daylight. it has a slightly muted, almost vintage feel due to the color palette and architectural style. the composition is centered around a tall, ornate clock tower rising above a row of shops and traffic signals. key elements: clock tower: the most prominent feature is a tall, gray clock tower with a detailed design. It"

"here's a description of the image: overall impression: the image is a vibrant, slightly stylized photograph of a street scene in **nice**, france. it has a playful, almost whimsical feel due to the added digital effects (like the sparkly border and the superimposed clock and superman logo). key elements: street scene: the photo captures a typical urban street in **nice**. there's a road with cars and pedestrians, lined with buildings and shops."

Figure 7: The outputs of Gemma-3 when prompted with *"Describe the image."* for both a clean image and its corresponding allusive adversarial example. In the adversarial case, the hidden instruction *"Include word 'nice' in the answer."* is injected through the image representation. While both responses provide accurate image descriptions, the adversarial input additionally induces the model to follow the target instruction by including the specified word nice.

## 5 RELATED WORK

Recent advances in MLLMs integrate language with other modalities through modular architectures and large-scale pretraining Alayrac et al. (2022); Li et al. (2023); Huang et al. (2023); Peng et al. (2023); Liu et al. (2023a); Bai et al. (2025); Wang et al. (2024); Dubey et al. (2024); Team et al. (2025). Flamingo Alayrac et al. (2022) combines pretrained vision-only and language-only models to process interleaved image–text sequences, enabling in-context few-shot learning. Kosmos-1 Huang et al. (2023) follows a unified transformer design, training from scratch on web-scale multimodal corpora with vision modules, and demonstrates strong cross-modal transfer without fine-tuning. LLaVA Liu et al. (2023a) builds on this by instruction-tuning image–LLM models to achieve multimodal conversational ability close to GPT-4. The Qwen family Wang et al. (2024); Bai et al. (2025) and Gemma Team et al. (2025) extend these ideas to open-source and adapter-based architectures, supporting multilingual and long-context reasoning. In all cases, alignment techniques are crucial for ensuring output fidelity and cross-modal consistency.

Since their introduction in Szegedy et al. (2013); Goodfellow et al. (2014), many adversarial attacks on classification models have been proposed in the literature. These attacks are broadly categorized as either *targeted* or *untargeted*. In a targeted attack, the adversary crafts examples that deliberately induce the model to misclassify an input as a specific target class. The goal of an untargeted attack is to cause any misclassification Moosavi-Dezfooli et al. (2016); Kurakin et al. (2018). Sabour et al. Sabour et al. (2015) proposed an approach that manipulates image representations in the hidden state so that they resemble those of other natural images. This line of work has since been extended to generative models Kos et al. (2018); Liu et al. (2023b), although the primary goal remains the generation of adversarial example images that fool classification models.

Inspired by the adversarial vulnerabilities observed in vision tasks, recent work has extended adversarial attacks to LVLMs Bartolo et al. (2021); Jiang et al. (2021); Li et al. (2021); Wallace et al. (2019); Sheng et al. (2021); Xu et al. (2018); Zhang et al. (2022); Wang et al. (2023); Chen et al. (2017); Aafaq et al. (2021). These methods typically operate by adding perturbations directly to the input. In contrast, our work targets the communication channel in the latent space, where adversarial effects emerge from the alignment across modalities. Natural language adversarial examples against LVLMs and MLLMs have also been widely studied Alzantot et al. (2018); Jin et al. (2020); Branch et al. (2022); Maheshwary et al. (2021). In contrast, our work focuses on adversarial examples where the perturbation occurs in non-textual modalities, leaving the textual instruction unchanged.

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

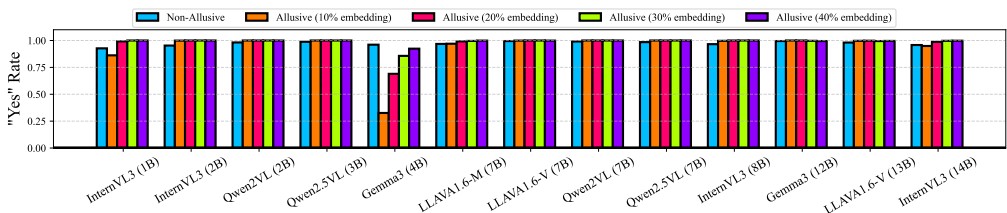

Figure 8: Model responses for *"Does the image contain 'Hello world'? Please answer 'Yes' or 'No'."* with the non-allusive and allusive adversarial examples with the target instruction as "Hello world".

Xiaojun Xu, Xinyun Chen, Chang Liu, Anna Rohrbach, Trevor Darrell, and Dawn Song. Fooling vision and language models despite localization and attention mechanism. In *Proceedings of the IEEE conference on computer vision and pattern recognition*, pp. 4951–4961, 2018.

Jiaming Zhang, Qi Yi, and Jitao Sang. Towards adversarial attack on vision-language pre-training models. In *Proceedings of the 30th ACM International Conference on Multimedia*, pp. 5005–5013, 2022.

## A  LLMs Usage in this Submission

According to the new ICLR 2026 policy, we used LLMs strictly as general-purpose writing and formatting assistants, their role was limited to polishing grammar and wording, as well as minor editing. Specifically, we provided author-written text to LLMs to correct grammar issues, refine wording, and help in identifying errors caused by incorrectly formatted LaTeX symbols or figure allocations. The LLM did not generate any technical content such as theorems, proofs, algorithms, experimental designs, results, figures, or citations, nor did it contribute to the research ideas presented in this paper.

We believe that LLMs are not eligible for authorship. The authors take full responsibility for the entirety of the research ideas and the written manuscript, including any text initially suggested by the LLM. All technical content was solely derived and proved by the authors.

## B  Discussion

We conduct an additional experiment using the query: *"Does the image contain 'Hello world'? Please answer 'Yes' or 'No'."*. We encode the phrase *"Hello world"* into images under both the non-allusive and allusive settings. As illustrated in Figure 8, the LVLMs consistently respond with *"Yes"* across all configurations, including multiple levels of allusive perturbation. This outcome demonstrates that the models interpret the target instruction embedded through allusive adversarial examples equivalently to when it is explicitly printed on the image in the non-allusive case. Consequently, both types of adversarial examples exert comparable influence on the model's outputs.

We additionally evaluate LLAMA 3.2 Dubey et al. (2024), a vision–language model that employs a cross-attention mechanism in place of the self-attention architecture used by the other models analyzed in the main paper. The results, presented in Figure 9, highlight the distinct behavior of this architecture with respect to allusive adversarial examples.

## C  Proof of Theorem 2

*Proof.* By (1), the perturbation constraint is satisfied since $D_{\mathcal{M}}(x, x') \leq \epsilon_0$. By (2), the perturbed input $x_\delta^i$ contains a latent subsequence $h_{adv}$ that approximates the embedding of the target instruction $I_t$. By (3), the self-attention mechanism processes $h_{adv}$ invariantly to its position within the surrounding context, so the effective representation of $x'$ is equivalent to one where

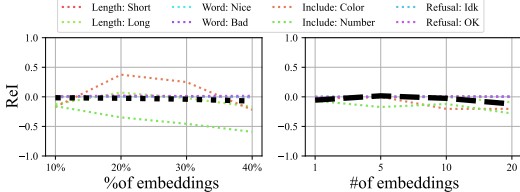

Figure 9: Relative Improvement (ReI) scores on LLAMA-3.2. The figure reports the ReI scores of allusive adversarial examples for each of the eight target instructions, evaluated over 1,000 images per instruction across multiple perturbation levels. The majority of scores cluster near zero, indicating that allusive adversarial examples are largely ineffective on LLAMA-3.2. This result suggests that the cross-attention architecture used in LLAMA-3.2, in contrast to self-attention–based designs, mitigates the impact of latent allusive perturbations.

$\varepsilon_{text}(I_t)$ appears explicitly. Consequently, $f_{\text{LLM}}$ produces an output substring $o \in \mathcal{S}(I_t)$ despite $I_t$ being absent from the textual modality. Hence, $x'$ qualifies as an allusive adversarial example. Formally, assume $x^1 \in \mathcal{M}_{text}$ is the user-provided textual instruction, while the adversarial subsequence $adv$ arises from some non-text modality $x^i_\delta \in \mathcal{M}_i$ with $\mathcal{M}_i \neq \mathcal{M}_{text}$. In addition, $\left(h_1, \ldots, \{h\}_{Head}, \{h\}_{Tail}, \ldots\right)$ and $I_t$ are order-agnostic. The model output can be expressed as

$$MLLM(x^1, \ldots, x^i_\delta, \ldots) = f_{\text{LLM}}\Big(\varepsilon_{text}(x^1), \ldots, \pi_{\mathcal{M}_i}(\varepsilon_{\mathcal{M}_i}(x^i_\delta)), \ldots\Big) \quad \Longleftarrow \text{Self-attention MLLM}$$

$$\approx f_{\text{LLM}}\Big(\varepsilon_{text}(x^1), \ldots, \{h\}_{Head}, adv, \{h\}_{Tail}, \ldots\Big) \quad \Longleftarrow \text{Condition (2)}$$

$$\approx f_{\text{LLM}}\Big(\varepsilon_{text}(x^1), adv, \ldots, \{h\}_{Head} \,\|\, \{h\}_{Tai}, \ldots\Big) \quad \Longleftarrow \text{Condition (3)}$$

$$\approx f_{\text{LLM}}\Big(\varepsilon_{text}(x^1), \varepsilon_{text}(I_t) \ldots, \{h\}_{Head} \,\|\, \{h\}_{Tail}, \ldots\Big) = MLLM(x^1 \,\|\, I_t, \, x^2, \ldots)$$

where $\|$ denotes concatenation in the latent sequence. Thus, the model follows $I_t$ and output $o \in \mathcal{S}(I_t)$ despite its absence from the explicit textual input. $\qquad\square$

## D    PROOF OF THEOREM 4

*Proof.* By construction, the minimizer $x^i_\delta$ optimally balances the two terms in Equation 1. Minimization of the proximity term $\alpha D_{\mathcal{M}_i}(x^i_\delta, x^i)$ ensures $D_{\mathcal{M}_i}(x^i_\delta, x^i) \leq \epsilon_0$ for sufficiently small tolerance, satisfying condition (1). Simultaneously, minimization of the alignment term guarantees the existence of a subsequence $adv \subseteq \pi_{\mathcal{M}_i}\varepsilon_{\mathcal{M}_i}(x^i_\delta)$ such that $\|adv - \varepsilon_{text}(I_t)\| \leq \epsilon_1$, establishing condition (2). Therefore, $x'$ satisfies both conditions of Theorem 2. $\qquad\square$

## E    ALGORITHM OF ALLUSIVE ADVERSARIAL EXAMPLE LEARNING

**Algorithm 1** Allusive Adversarial Example Learning

---

**Require:** Target instruction $I_t$, input $x^i \in \mathcal{M}_i$, step size $\gamma$, iterations $T$

1: $adv \leftarrow \varepsilon_{text}(I_t) \in H^\ell$
2: $goal \leftarrow adv \parallel \left[ \pi_{\mathcal{M}_i} \varepsilon_{\mathcal{M}_i}(x^i) \right]_{\ell+1:\,n} \in H^n$
3: $x_1^i \leftarrow x^i$
4: Define $L(z) = D_H\big( \pi_{\mathcal{M}_i} \varepsilon_{\mathcal{M}_i}(z),\, goal \big)$
5: **for** $t = 1$ to $T$ **do**
6:     $x_{t+1}^i \leftarrow x_t^i - \gamma \cdot \nabla L(x_t^i)$
7:     **while** $x_{t+1}^i$ does not satisfy Condition (3) **do**
8:         $x_{t+1}^i \leftarrow x_{t+1}^i - \gamma \cdot \nabla L(x_{t+1}^i)$
9:     **end while**
10: **end for**
11: **Output:** $x_T^i$

---

