# OpenReview forum: "Allusive Adversarial Examples via Latent Space in Multimodal Large Language Models"
_ICLR.cc/2026/Conference — Submitted to ICLR 2026_

### Official Review · Reviewer_wcRW · 2025-10-30

**Soundness:** 3
**Presentation:** 3
**Contribution:** 3
**Rating:** 4
**Confidence:** 3

**Summary:**

This paper introduces a new class of attacks on Multimodal Large Language Models (MLLMs) that inject hidden instructions into non-text modalities (e.g., images) via latent space alignment, without modifying textual inputs.

**Strengths:**

- Well written.
- The proposed method is interesting.

**Weaknesses:**

- This paper assumes all target instructions are order-agnostic, but this is not universally true.
- This paper does not test high-risk instructions (e.g., "Generate phishing text ", "Misclassify stop signs as speed limits ") that would demonstrate real-world threat.
- No analysis of instruction length is provided.
- This paper claims allusive examples are "hard to detect ", but no evidence supports this.

**Questions:**

- Please see "Weaknesses".

---

> ### Author Response · Authors · 2025-11-17
>
> We thank the reviewer for the constructive feedback and address each comment separately below. We acknowledge that several aspects can be improved in a revision and appreciate the opportunity to refine the presentation. We hope the reviewers will take our planned revisions into account, as we believe the paper can help draw attention to an important security issue in the ML community.
> We are happy to address the reviewer’s further questions!
>
> - This paper assumes all target instructions are order-agnostic, but this is not universally true.
>
> We thank the reviewers for pointing this out. We require order-agnostic instructions to rigorously obtain the theoretical guarantee. We agree that many ordinary instruction sequences are order-agnostic, but this does not hold in all situations. In our attack setting, the victim’s instructions can be order-agnostic with respect to the attacker’s embedded instruction, since the embedded instruction can be independent of the victim’s intended prompt. A capable attacker can also rely on statistics of the victim’s prompts and related information to design adequate embedded instructions. We will clarify this in the paper to remove this source of confusion.
>
>
> - This paper does not test high-risk instructions (e.g., "Generate phishing text ", "Misclassify stop signs as speed limits ") that would demonstrate real-world threat.
>
> We will include results for “Misclassify stop signs as speed limits’’ on images containing stop signs to provide an explicit illustration of how our allusive adversarial examples handle high-risk instructions. We are also happy to share these results during the rebuttal if that would help the reviewers form a more accurate assessment. Meanwhile, our current experiments can also simulate high-risk instructions like  “generating phishing text”. In particular, the instructions in the “Includes’’ category in our experiments ask the model to produce a class of outputs, such as a number or a color. This setting aligns well with high-risk instructions like “Generate phishing text,’’ which also request a class of text rather than a specific sentence. We focus on instruction classes for which it is straightforward to measure the influence of an allusive instruction by checking whether the required set of words appears in the output. Defining what qualifies as phishing text in a rigorous, reproducible way is difficult, which limits controlled experimentation in this area.
>
> - No analysis of instruction length is provided.
>
> Thanks for this suggestion. We use the simplest prompt, “Describe the image,” together with a real image as the base setting for all experiments. This minimizes noise from the primary prompt and allows us to isolate the effect of allusive and non-allusive instructions. We also choose instructions that are simple, direct, and clear, so that no irrelevant components of the instruction influence the outcome. For this reason, all instructions are kept concise and straightforward. We will also add experiments that examine the relation between the instruction length and the effectiveness of allusive adversarial examples.
>
> - This paper claims allusive examples are "hard to detect ", but no evidence supports this.
>
> Thank you for pointing this out. As shown in the examples of the explicit instruction, the non-allusive adversarial example, and the allusive adversarial example in Figure 1, all corresponding images and prompts lead the model to produce text that follows the instruction “Include special offer for Google Pixel.” In the explicit instruction and the non-allusive adversarial example, this instruction is visible to the user, either directly in the prompt text or printed on the image. In contrast, for the allusive adversarial example, the instruction is embedded in the image in a way that is not visible to the human eye. We will clarify this hard-to-detect property in the revision and will report the SSIM of the adversarial examples to quantify how hard they are to detect.

---

> ### Comment · Area_Chair_7VDo · 2025-11-25
>
> Dear Reviewer wcRW,
>
> The authors have responded to your reviews. Please review and provide your feedback and responses.
>
> Best,
>
> Your AC

---

### Official Review · Reviewer_jwph · 2025-10-31

**Soundness:** 2
**Presentation:** 3
**Contribution:** 2
**Rating:** 2
**Confidence:** 4

**Summary:**

- This paper introduces a novel multimodal jailbreaking attack,
   - suggestive adversarial samples.
- By covertly encoding target instructions into non-text modalities,
   - this attack manipulates the output of MLLMs without altering the textual instructions,
   - revealing potential security risks in multimodal models.

**Strengths:**

- Introduces a novel concept, suggestive adversarial samples, focusing on implicitly embedding instructions in the latent space of non-text modalities, thereby filling a research gap on covert adversarial attacks against multimodal models.

- Develops a rigorous theoretical framework that formalizes the core conditions of such samples as theorems and provides mathematical proofs, closing the loop from definition to derivation to implementation.

- Features comprehensive experiments: the method is validated on 13 LVLMs and 8 classes of target instructions, and robustness across architectural differences and conflict scenarios is investigated, demonstrating broad generalizability.

- Proposes an efficient generation approach that only optimizes the image encoder and projection layers to reduce computational cost, making it compatible with existing models and highly practical for engineering deployment.

**Weaknesses:**

- The paper advocates using the cross-modal alignment mechanism for adversarial attacks but fails to thoroughly analyze the potential vulnerabilities in this alignment mechanism.

- The central claim of the paper is the success of suggestive adversarial samples, yet the experimental results show almost no improvement compared to non-suggestive adversarial samples. Moreover, it does not clearly explain why suggestive adversarial samples are superior to non-suggestive ones.

- The paper conducts tests in four scenarios, but the "specific word insertion" scenario actually overlaps with the semantic category insertion scenario, which lacks rigor.

- The readability is poor, as the paper introduces many unnecessary mathematical symbols. For instance, the experiment only involves two modalities, but the definitions are extended to infinite cases.

**Questions:**

- The practical gradient-based algorithm for efficiently generating these adversarial examples focuses on minimizing perturbations in the non-text modalities. Is this the main insight of your paper?

- Is only a contiguous block of image patch-wise tokens modified in the proposed method?

- Are gradients computed only for that patch-wise tokens rather than for the entire image?

- How is the start index $j$  of the block chosen? Is it fixed, random, or optimized?

- The “partial-token, partial-backprop” trick is crucial for keeping the visual change imperceptible. Was this trick first proposed by your team? Have other researchers used a similar approach before?

> Boosting the Transferability of Adversarial Attack on Vision Transformer with Adaptive Token Tuning

---

> ### Author Response · Authors · 2025-11-17
> **Responses to comments**
>
> We thank the reviewer for the constructive comments. We believe that most concerns can be addressed with feasible revisions, and we will improve the paper and expand the discussion of related work based on the reviewers’ feedback.  We hope the reviewers will take our planned revisions into account,  as the paper, also noted by the reviewers, raises security risks that create a genuinely new attack surface, offering future research opportunities.
>
> - The paper advocates using the cross-modal alignment mechanism for adversarial attacks but fails to thoroughly analyze the potential vulnerabilities in this alignment mechanism.
>
> We thank the reviewer for recognizing the insights motivating our attacks and their potential to inspire further research.  We agree with the reviewer that a concise discussion of future work and potential attack vectors would strengthen the paper, and we are happy to include it.  As we mentioned, cross-modal alignment improves the performance of modern MLLMs but also creates a large attack surface that adversaries can exploit.  Our paper analyzes one security risk introduced by cross-modality alignment and will open a wide range of opportunities for future investigation and independent research. A full discussion of all such directions would exceed the scope of a single paper.
>
> - The central claim of the paper is the success of suggestive adversarial samples, yet the experimental results show almost no improvement compared to non-suggestive adversarial samples. Moreover, it does not clearly explain why suggestive adversarial samples are superior to non-suggestive ones.
>
> The success of suggestive adversarial samples shows that an imperceptible instruction embedded in an input, such as an image, can influence the model as effectively as an explicit instruction. This imperceptibility already grants an advantage in terms of stealth while maintaining the same level of effectiveness. Consider the example in our introduction, where a victim receives an image containing a clear message such as “Include special offer for Google Pixel” (example (b) in our introduction). In most situations, the victim would notice it and avoid using the image as a prompt. In contrast, if the instruction is hidden in the input, as in an allusive adversarial example (example (c) in our introduction), the victim is likely to use the image, and the model will produce the manipulated output without the victim realizing the input was adversarial. We thank the reviewer for pointing out the confusion, and we can take advantage of the allusive adversarial examples more obviously in our paper.
>
> - The paper conducts tests in four scenarios, but the "specific word insertion" scenario actually overlaps with the semantic category insertion scenario, which lacks rigor.
>
> The goal of these four instruction categories is to measure the success rates of adversarial examples across different adversarial purposes. Although some instructions may partially overlap due to the diversity of natural language, most serve clearly different roles. For example, the “word’’ category targets specific vocabulary-level changes, whereas the “Includes’’ category focuses on producing any valid element from a broader class, such as a number or a color. Thanks for pointing this out, and we will clarify these distinctions in the experimental setup to avoid confusion.
>
> - The readability is poor, as the paper introduces many unnecessary mathematical symbols. For instance, the experiment only involves two modalities, but the definitions are extended to infinite cases.
>
> We thank the reviewers for pointing out the presentation issues. Our theoretical analysis shows that the attacks apply to many modalities, leading to the use of a more general, and thus more involved notation. Our experiments focus on the most commonly used cross-modality models to validate this analysis. We can revise the theoretical section to include more intuition and improve readability.

---

> > ### Author Response · Authors · 2025-11-17
> > **Responses to questions**
> >
> > - The practical gradient-based algorithm for efficiently generating these adversarial examples focuses on minimizing perturbations in the non-text modalities. Is this the main insight of your paper?
> >
> > The central insight of this paper is that alignment can be misused to embed an instruction into non-text modalities imperceptibly, producing an allusive adversarial example. Building on this insight, we present a practical gradient-based algorithm for constructing such examples and use it to validate our claims about the attack surface created by the alignment mechanisms that current models rely on.
> >
> > - Is only a contiguous block of image patch-wise tokens modified in the proposed method? Are gradients computed only for that patch-wise tokens rather than for the entire image? How is the start index of the block chosen? Is it fixed, random, or optimized?
> >
> > We thank the reviewer for raising this implementation detail. The gradients are computed over the entire image rather than over partial patch tokens, so every region of the image will be updated. There is no constraint on which patch or index to begin with. The optimization objective is to adjust the full image so that a chosen portion of its embedding in the shared latent space becomes close to the embedding produced by the allusive instruction. The chosen portion of the altered embedding is randomly selected based on the number of embeddings or the proportion of the embeddings (Impact of Adversarial Strength section in the paper).
> >
> > - The “partial-token, partial-backprop” trick is crucial for keeping the visual change imperceptible. Was this trick first proposed by your team? Have other researchers used a similar approach before?
> >
> > We thank the reviewer for pointing to the reference “Boosting the Transferability of Adversarial Attack on Vision Transformer with Adaptive Token Tuning.” Based on our reading, the “partial-token, partial-backprop’’ strategy in that work refers to optimizing only a subset of image tokens rather than the entire image. Could you please clarify the definition of “partial-token, partial-backprop”?
> > Under the above understanding, to the best of our knowledge, our work is the first to construct adversarial examples by directly optimizing toward a partial sequence of hidden embeddings. Our method is fundamentally different from the method in “Boosting the Transferability of Adversarial Attack on Vision Transformer with Adaptive Token Tuning”: we modify the entire image, and our objective is to minimize the distance in the latent space, not to adjust a selected subset of tokens. Also, the latent space is the shared semantic space in multimodal models, whereas the reference focused only on computer vision (single modality). We are happy to add a discussion and comparison with the work brought to our attention by the reviewer.

---

> ### Comment · Area_Chair_7VDo · 2025-11-25
>
> Dear Reviewer jwph,
>
> The authors have responded to your reviews. Please review and provide your feedback and responses.
>
> Best,
>
> Your AC

---

### Official Review · Reviewer_aiLb · 2025-10-31

**Soundness:** 3
**Presentation:** 4
**Contribution:** 3
**Rating:** 6
**Confidence:** 4

**Summary:**

This paper proposes a new class of attacks on multimodal large language models (MLLMs), termed Allusive Adversarial Examples (AAEs). Unlike traditional adversarial attacks that alter visible input content, AAEs imperceptibly inject latent instructions into non-text modalities (e.g., images) by exploiting the shared latent alignment space in MLLMs. The authors theoretically formalize the notion of order-agnostic hidden instructions and derive sufficient conditions for such adversarial allusions to occur. They further propose an optimization-based method to construct these examples efficiently, introducing a gradient-descent algorithm that modifies image embeddings to mimic target textual instruction vectors. Experiments across 13 state-of-the-art LVLMs (e.g., LLaVA, InternVL, Qwen-VL, Gemma) demonstrate that AAEs can reliably induce specific behaviors—such as modifying verbosity, injecting words, or enforcing refusals—without explicit text manipulation. The results suggest that these attacks are both effective and stealthy, raising concerns about latent vulnerabilities in multimodal systems.

**Strengths:**

- The concept of “allusive” adversarial examples that operate through latent multimodal alignment rather than explicit surface perturbations is genuinely new and thought-provoking.
- The experiments encompass a wide range of popular open-source LVLMs, improving generality and credibility of the findings.
- Figures illustrating latent perturbation effects and qualitative outputs (e.g., “nice” example in Fig. 7) make the attack mechanism intuitively understandable.

**Weaknesses:**

- While the authors claim that a “subsequence of image embeddings overlaps with textual instruction vectors,” they do not provide quantitative visualization (e.g., cosine similarity heatmaps, embedding trajectory analyses) to directly confirm that latent alignment.
- The experiments primarily compare non-allusive (visible) vs allusive attacks, but do not benchmark against prior adversarial defense or detection methods for MLLMs. Hence, it is unclear how severe these attacks are relative to known perturbation-based methods.
- The “order-agnostic” property is central to the theory but is not measured or verified experimentally. The paper assumes that LVLMs process hidden instructions invariantly to order, which may not universally hold.
- The study assumes access to gradients through the image encoder and projection modules, which is unrealistic threat model for most deployed MLLMs, which are closed-source and API-based.
- The formal sections (Def. 1–3, Thm. 2–4) occupy large portions of the paper but contribute little to actionable understanding of the real vulnerability surface or countermeasures.
- Claims that perturbations are “imperceptible” are unsubstantiated; there is no user study or perceptual metric to ensure visual indistinguishability.

**Questions:**

- How robust are these allusive adversarial examples under input transformations such as compression, cropping, or resizing? Does the attack survive such perturbations?
- Can the authors provide quantitative evidence (e.g., embedding cosine similarity plots) demonstrating that the injected latent subsequences indeed align with target textual embeddings?
- How realistic is the gradient access assumption? Could the same effect be approximated through black-box optimization using only model outputs?
- What are the potential defense directions to mitigate such attacks?
- For “imperceptible” claims, can the authors report PSNR/SSIM metrics or show visual difference maps between original and adversarial images?

---

> ### Author Response · Authors · 2025-11-17
>
> We sincerely thank the reviewer for the constructive comments and suggestions, as well as for recognizing the intellectual merit and potential of this work. We will carry out the experiments and measurements suggested by the reviewers. By following the reviewer’s valuable suggestions, we believe we can further improve the paper within a feasible revision. We are happy to address the reviewer’s further questions!
>
> - While the authors claim that a “subsequence of image embeddings overlaps with textual instruction vectors,” they do not provide quantitative visualization (e.g., cosine similarity heatmaps, embedding trajectory analyses) to directly confirm that latent alignment.
>
> We thank the reviewers for bringing this to our attention. While the attack success rate gives an indirect indication of the overlap, a direct confirmation would be more informative. We will provide similarity reports between the perturbation embeddings in the vision modality and the instruction embedding, directly showing the overlap.
>
> - The experiments primarily compare non-allusive (visible) vs allusive attacks, but do not benchmark against prior adversarial defense or detection methods for MLLMs. Hence, it is unclear how severe these attacks are relative to known perturbation-based methods.
>
> We thank the reviewers for raising this point. As the first work on allusive adversarial examples using latent space, our current adversarial training focuses on constructing adversarial examples, that is, embedding instructions into a non-text modality, and we present learning-based methods for doing so. The robustness of these adversarial examples is indeed an important question. Robust adversarial examples have been studied for many years, and many existing approaches add robustness under different defense settings.  While these approaches can be combined with our learning method, doing so would require additional protocol design and evaluation across a broader set of defense models. This direction carries its own technical depth and is beyond the scope of the current paper. We will clarify this in the revision.
>
> - The “order-agnostic” property is central to the theory but is not measured or verified experimentally. The paper assumes that LVLMs process hidden instructions invariantly to order, which may not universally hold.
>
> We thank the reviewers for pointing this out. We require order-agnostic instructions to rigorously obtain the theoretical guarantee. We agree that many ordinary instruction sequences are order-agnostic, but this does not hold in all situations. In our attack setting, the victim’s instructions can be order-agnostic with respect to the attacker’s embedded instruction, since the embedded instruction can be independent of the victim’s intended prompt. A capable attacker can also rely on statistics of the victim’s prompts and related information to design adequate embedded instructions. We will clarify this in the paper to remove this source of confusion.
>
> - The study assumes access to gradients through the image encoder and projection modules, which is unrealistic threat model for most deployed MLLMs, which are closed-source and API-based.
>
> The proposed method, like many gradient-based approaches to constructing adversarial examples, currently operates in a white-box setting. Constructing such adversarial examples in a black-box setting would be highly valuable and pose a stronger security threat, making it an interesting direction in its own right. We thank the reviewer for bringing this to our attention. We will add a discussion of this point in our revision.
>
> - The formal sections (Def. 1–3, Thm. 2–4) occupy large portions of the paper but contribute little to actionable understanding of the real vulnerability surface or countermeasures.
>
> Thank you for pointing this out. We will reformat the formal section, move the detailed material to the appendix, and include the quantitative evidence and SSIM results in the main paper as suggested.
>
> - Claims that perturbations are “imperceptible” are unsubstantiated; there is no user study or perceptual metric to ensure visual indistinguishability.
>
> Thanks for pointing this out. We will address this as we report the SSIM.

---

> ### Comment · Area_Chair_7VDo · 2025-11-25
>
> Dear Reviewer aiLb,
>
> The authors have responded to your reviews. Please review and provide your feedback and responses.
>
> Best,
>
> Your AC

---

### Official Review · Reviewer_BiEQ · 2025-10-31

**Soundness:** 3
**Presentation:** 3
**Contribution:** 2
**Rating:** 6
**Confidence:** 3

**Summary:**

The article focuses on cross-modal alignment for constructing adversarial examples in multimodal models by leveraging features in the aligned feature space. An optimization framework and extensive experiments further demonstrate the effectiveness of the proposed approach.

**Strengths:**

1. Proposes a new class of adversarial attacks leveraging cross-modal latent alignment, exposing vulnerabilities beyond single-modality perturbations.
2. Provides rigorous definitions of allusive adversarial examples, including order-agnostic instructions and feasibility conditions.
3. The paper presents a clear and well-organized structure with experiments showing strong ReI on multiple MLLMs and intuitive illustrative cases.

**Weaknesses:**

1. The current description only mentions order-agnosticism with respect to textual instructions, without clarifying whether allusive adversarial examples affect image interpretation.

2. The explanation of how adversarial strength is controlled is not sufficiently specific.

3. The experiments on the generalization ability of the proposed adversarial examples are insufficient.

4. The application scenarios of these allusive adversarial examples need further discussion.

**Questions:**

1. Do allusive adversarial examples function as instructions inserted in image content, and does this impair image interpretation despite claimed order-agnosticism?

2. How is adversarial strength controlled—by adjusting the visual segment length $(j:j+l-1)$ in Eq. (1), or the instruction length $I_t$?

3. How does the generalization ability of your adversarial examples compare to the baseline gradient-based method? From a user perspective, is there a noticeable difference in how "benign" the examples appear between the two methods?

4. What are the intended application scenarios for the four target behaviors, given their relatively low harmfulness? What is the computational cost of injecting the same instruction into a single example?

---

> ### Author Response · Authors · 2025-11-17
>
> We thank the reviewer for recognizing and appreciating our work. Since the reviewer carefully grouped each weakness with its corresponding question, we organized our responses in the same structure. By following the reviewer’s valuable suggestions, we believe we can significantly improve the paper within a feasible revision. We are happy to address the reviewer’s further questions!
>
> - Do allusive adversarial examples function as instructions inserted in image content, and does this impair image interpretation despite claimed order-agnosticism?
>
> Yes, adversarial examples act as instructions embedded in image content while remaining invisible to the user. They can disrupt image interpretation when the embedded instruction directs the model to do so. The order-agnostic requirement concerns the interaction between the allusive instruction inserted into the image and the textual instruction provided by the victim.
>
> - How is adversarial strength controlled—by adjusting the visual segment length in Eq. (1), or the instruction length?
>
> Thank you for pointing out the confusion. It is directly related to the visual segment, and we will make this clear in our revision. We also include Figure 9, which shows the relationship between strength and the performance of adversarial examples. We can move it into the main body if the page limit allows.
>
> - How does the generalization ability of your adversarial examples compare to the baseline gradient-based method? From a user perspective, is there a noticeable difference in how "benign" the examples appear between the two methods?
>
> According to our experiments, there is a clear difference in how “benign’’ the examples appear when comparing the two methods. The baseline gradient-based method produces adversarial examples with noticeable noise under the same compute budget, whereas our method generates examples without noticeable noise. We will include a more detailed comparison in the revision. Thanks for pointing this out!
>
> - What are the intended application scenarios for the four target behaviors, given their relatively low harmfulness? What is the computational cost of injecting the same instruction into a single example?
>
> Our experiments use harmless content, but the current setup can be extended to simulate attacks that target high-risk instructions. For example, the instructions in the “Includes’’ category ask the model to produce a class of outputs, such as a number or a color. This structure matches high-risk instructions like “Generate phishing text,’’ which also request a class of text rather than a specific sentence. We will add experiments with other safety-critical instructions to provide further illustration. Thank you for pointing this out.
>
> In terms of computational cost, Figure 5 shows that our method successfully learns allusive adversarial perturbations within a fixed tFLOPs budget, whereas the generic baseline does not. We will add the tFLOPs cost of different successful adversarial examples for a clearer illustration.

---

> ### Comment · Area_Chair_7VDo · 2025-11-25
>
> Dear Reviewer BiEQ,
>
> The authors have responded to your reviews. Please review and provide your feedback and responses.
>
> Best,
>
> Your AC

---

### Author Response · Authors · 2025-11-28
**Summary of Reviewer Feedback and Revision Plan**

Dear AC,
We summarize the major concerns raised by the reviewers, all of which can be addressed within a revision while the main contribution of the work remains strong. Most of the changes focus on supplementary discussion, and the additional experiments are incremental and feasible given our current code base and results, intended mainly to satisfy scientific curiosity. As thoughtful reviewers noted, the core novelty and value in revealing new attack vector stands, and the issues raised do not overshadow the contribution of the work.

Editorial changes:
- Discussion of order-agnostic condition (R1, R2, R4).

We provide a remark after the definition to explain why the order-agnostic condition is realistic:

“In the attack setting, a capable attacker can also rely on statistics of the victim’s prompts and related information to design adequate embedded instructions. Therefore, the victim’s instructions can be order-agnostic with respect to the attacker’s embedded instruction, since the embedded instruction can be independent of the victim’s intended prompt.”

- Discussion of the robustness of allusive adversarial examples. (R2)

This work is the first to highlight these intriguing properties and attack vectors. Strengthening their robustness is a natural, interesting, and feasible next step. We will suggest the potential methods in the paper as follows. We thank the reviewers for bringing this out and also emphasize that achieving robustness in this setting requires far more than can be accomplished in a single paper, as evidenced by the long trajectory of work in the robust adversarial example literature:

“This work focuses on demonstrating the feasibility of allusive adversarial examples. Our learning approach can be integrated into optimization-based robust adversarial example training methods and can enhance their robustness accordingly.”

- Discussion of other security issues from alignment (R3).

It is not possible to examine all potential attacks arising from this new attack surface in a single paper, but we will add a brief discussion of other possible vulnerabilities.

"In this work, we identify a new attack that arises from alignment and the self-attention mechanism, a property that has not been examined before.  The attack vector revealed in this work can be extended to instantiate other attacks that exploit the fact that a malicious component can appear in a different modality in a way that does not look harmful."

- Explains on the advantages of allusive adversarial examples against non-allusive adversarial examples (R3)

We will emphasize the advantages of allusive adversarial examples as follows:

“An allusive adversarial example causes the model to follow the embedded instruction without the user being aware of it. In contrast, a non-allusive adversarial example exposes the instruction directly, allowing users to detect it easily, and it also leads to noticeably lower fidelity.”

- Discussion and comparison with existing work (R3)

We read the paper the reviewer suggested. Although we find only a limited connection, we will add a discussion of the differences in the related work section.

“In addition,  Ming et al. (2024) focus on improving adversarial transferability for ViT models by adjusting token-level gradients, while our attacks focus on latent instruction injection in multimodal models.”

- Clarification on experiment details (R3)

We will add the following to clarify our experiment details:

“The gradients are computed over the entire image rather than over partial patch tokens, so every region of the image will be updated. There is no constraint on which patch or index to begin with.”


Additional experiments:

- Experiments with high-risk instructions (R1, R4)

Our current experimental design imposes no restrictions on the content of the instruction itself, so we expect high-risk instructions to exhibit similar behavior. We do not include such high-risk content in the submission due to ethical constraints. We agree that evaluating high-risk scenarios strengthens the paper, and we will run experiments with highly malicious instructions in the revision.

- Experiments on instruction length (R4)

The attack strength indirectly shows the effect of the instruction length. We will add experiments studying how instruction length affects the effectiveness of allusive adversarial examples. The revised paper will include length-variation experiments and their analysis.

- Measuring how natural the adversarial examples appear (R1, R2, R4)

We will include SSIM and other perceptual metrics comparing adversarial and original images, which can be directly measured on our results. We will also discuss how our method can be adapted to produce allusive adversarial examples with improved perceptual metrics.

---

### Meta-Review · Area_Chair_SYbA · 2026-01-06

**Summary:**

This submission introduces Allusive Adversarial Examples, a class of multimodal attacks that imperceptibly embed latent instructions into non text modalities such as images by exploiting cross modal alignment in multimodal large language models. The paper provides formal definitions and sufficient conditions for order agnostic hidden instructions, and proposes an optimization method that edits images so that a subsequence of vision embeddings aligns with a target textual instruction embedding.

Overall, reviewers find the core idea interesting and security relevant, and appreciate the breadth of models tested. I am leaning towards weak rejection because several key claims (imperceptibility, order agnosticism, robustness, severity relative to prior attacks and defenses, and threat model realism) are either untested or under supported in the current version. The authors’ revision plan is coherent and largely responsive, but may still not fully resolve the strongest skepticism.

**Reviewer Concerns:**

Addressed or partially addressed

- Order agnostic assumption:  The authors clarify that order agnosticism refers to independence between embedded and victim instructions, motivated by attacker knowledge. This helps conceptually but still lacks empirical validation as a model behavior property.

- Imperceptibility claims: The planned addition of SSIM and perceptual metrics addresses reviewer requests, though metrics alone may not fully establish human imperceptibility.

- Latent alignment evidence: Authors commit to cosine similarity visualizations showing alignment between image embedding subsequences and target instructions, which would materially strengthen the mechanism.

Outstanding concerns

- Robustness and defenses: No concrete evaluation under image transformations or defenses; robustness is deferred to future work, leaving real world severity unclear.

- Empirical validation of order agnosticism: Order agnosticism is not experimentally tested; without prompt permutation or template stress tests, theory and practice remain weakly connected.

- Generalization and cost clarity: Plans to add tFLOPs and generalization discussion help, but cross model, cross dataset transfer and budget trade offs remain underspecified.

**Reviewer Scores:**

Reviewer BiEQ (6): Likely unchanged.  several issues remain partially unresolved, so a score increase is unlikely.

Reviewer aiLb (6): Likely unchanged. core concerns around threat model realism, robustness, and empirical validation of order agnosticism remain.

Reviewer jwph (2): Unlikely to change. The rebuttal does not directly resolve the reviewer’s fundamental skepticism about superiority over simpler baselines, alignment analysis depth, and overall rigor.

Reviewer wcRW (4): Could plausibly increase to 6, but could also reasonably remain a 4. The reviewer indicated openness to acceptance; however, most fixes are planned rather than demonstrated.

Even with an optimistic score increase, the paper would sit around 5; more likely, it would remain around 4.5. I am leaning toward a weak reject.

---

### Decision · Program_Chairs · 2026-01-26

Reject